# RETHINKING CONVOLUTION: TOWARDS AN OPTIMAL EFFICIENCY

## ABSTRACT

In this paper, we present our recent research about the computational efficiency in convolution. Convolution operation is the most critical component in recent surge of deep learning research. Conventional 2D convolution takes $O(C^2 K^2 HW)$ to calculate, where $C$ is the channel size, $K$ is the kernel size, while $H$ and $W$ are the output height and width. Such computation has become really costly considering that these parameters increased over the past few years to meet the needs of demanding applications. Among various implementations of the convolution, separable convolution has been proven to be more efficient in reducing the computational demand. For example, depth separable convolution reduces the complexity to $O(CHW \cdot (C + K^2))$ while spatial separable convolution reduces the complexity to $O(C^2 KHW)$. However, these are considered an ad hoc design which cannot ensure that they can in general achieve optimal separation. In this research, we propose a novel operator called *optimal separable convolution* which can be calculated at $O(C^{\frac{3}{2}} KHW)$ by optimal design for the internal number of groups and kernel sizes for general separable convolutions. When there is no restriction in the number of separated convolutions, an even lower complexity at $O(CHW \cdot \log(CK^2))$ can be achieved. Experimental results demonstrate that the proposed optimal separable convolution is able to achieve an improved accuracy-FLOPs and accuracy-#Params trade-offs over both conventional and depth/spatial separable convolutions.

## 1 INTRODUCTION

Tremendous progresses have been made in recent years towards more accurate image analysis tasks, such as image classification, with deep convolutional neural networks (DCNNs) (Krizhevsky et al., 2012; Srivastava et al., 2015; He et al., 2016; Real et al., 2019; Tan & Le, 2019; Dai et al., 2020). However, the computational complexity for state-of-the-art DCNN models has also become increasingly high and computationally expensive. This can significantly defer their deployment to real-world applications, such as mobile platforms and robotics, where the resources are highly constrained (Howard et al., 2017; Dai et al., 2020). It is very much desired that a DCNN could achieve better performance with less computation and fewer model parameters.

The most time-consuming building block of a DCNN is the convolutional layer. There have been many previous works aiming at reducing the amount of computation in the convolutional layer. Historically, researchers apply Fast Fourier Transform (FFT) (Nussbaumer, 1981; Quarteroni et al., 2010) to implement convolution and they gain great speed up for large convolutional kernels. For small convolutional kernels, a direct application is often still cheaper (Podlozhnyuk, 2007). Researchers also explore low rank approximation (Jaderberg et al., 2014; Ioannou et al., 2015) to implement convolutions. However, most of the existing methods start from a pre-trained model and mainly focus on network pruning and compression.

In this research, we study how to design a separable convolution to achieve an optimal implementation in terms of computational complexity. Enabling convolution separable has been proven to be an efficient way to reduce the computational complexity (Sifre & Mallat, 2014; Howard et al., 2017; Szegedy et al., 2016). Comparing to the FFT and low rank approximation approaches, a well-designed separable convolution shall be efficient for both small and large kernel sizes and shall not require a pre-trained model to operate on.

Table 1: A comparison of computational complexity and the number of parameters of the proposed optimal separable convolution and existing approaches. The proposed optimal separable convolution is much more efficient. In this table, $C$ represents the channel size of convolution, $K$ is the kernel size, $H$ and $W$ are the output height and width, $g$ is the number of groups. "Vol. RF" represents whether the corresponding convolution satisfies the proposed volumetric receptive field condition.

| | Conventional Conv2D | Grouped Conv2D | Depth-wise Conv2D | Point-wise Conv2D | Depth Separable Conv2D | Spatial Separable Conv2D | Optimal Separable Conv2D ($N = 2$) | Optimal Separable Conv2D (Optimized $N$) |
|---|---|---|---|---|---|---|---|---|
| FLOPs | $C^2 K^2 HW$ | $C^2 K^2 HW/g$ | $CK^2 HW$ | $C^2 HW$ | $CHW(C + K^2)$ | $2C^2 KHW$ | $2C^{\frac{3}{2}} KHW$ | $eCHW \log(CK^2)$ |
| #Params | $C^2 K^2$ | $C^2 K^2/g$ | $CK^2$ | $C^2$ | $C(C + K^2)$ | $2C^2 K$ | $2C^{\frac{3}{2}} K$ | $eC \log(CK^2)$ |
| Vol. RF | ✓ | ✗ | ✗ | ✗ | ✓ | ✓ | ✓ | ✓ |
| Note | – | – | $g = C$ | $K = 1$ | Depth-wise + Point-wise | $K^2 \to 2K$ | – | $e = 2.71828\ldots$ |

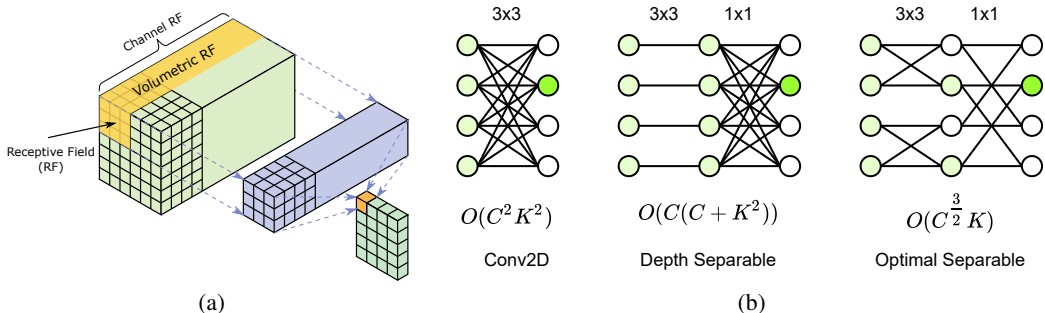

(a)            (b)

Figure 1: Volumetric receptive field and the proposed optimal separable convolution. (a) The volumetric receptive field (RF) of a convolution is the Cartesian product of its (spatial) RF and channel RF. (b) Illustrations of the channel connections for conventional, depth separable, and the proposed optimal separable convolutions.

In the DCNN research, two most well-known separable convolutions are depth separable (Sifre & Mallat, 2014) and spatial separable (Szegedy et al., 2016) convolutions. Both are able to reduce the computational complexity of a convolution. The complexity of a conventional 2D convolution is quadratic with three hyper-parameters: number of channels ($C$), kernel size ($K$), and spatial dimensions ($H$ or $W$), and its computational complexity is actually $O(C^2 K^2 HW)$. Depth separable convolution is constructed as a depth-wise convolution followed by a point-wise convolution, where depth-wise convolution is a group convolution with its number of groups $g = C$ and point-wise convolution is a $1 \times 1$ convolution. Spatial separable convolution replaces a $K \times K$ kernel to $K \times 1$ and $1 \times K$ kernels. Different types of convolutions and their computational costs are summarized in Table 1. From this table, we can easily verify that depth separable convolution has a complexity of $O(CHW \cdot (C + K^2))$ and spatial separable convolution has a complexity of $O(C^2 KHW)$.

Both depth separable and spatial separable convolutions follow an ad hoc design. They are able to reduce the computational cost to some degree but normally will not achieve an optimal separation. A separable convolution in general has three sets of parameters: the internal number of groups, channel size, and kernel size of each separated convolution. Instead of setting these parameters in an ad hoc fashion, we design a scheme to achieve an optimal separation. The resulting separable convolution is called *optimal separable convolution* in this research.

To prevent the proposed optimal separable convolution from being degenerated, we assume that the internal channel size is in an order of $O(C)$ and propose the following *volumetric receptive field condition*. As illustrated in Fig. 1a, similar to the *receptive field (RF)* of a convolution which is defined as the region in the input space that a particular CNN's feature is looking at (or affected by) (Lindeberg, 2013), we define the *volumetric RF* of a convolution to be the volume in the input space that affects CNN's output. The volumetric RF condition requires that a properly decomposed separable convolution maintains the same volumetric RF as the original convolution before decomposition. Hence, the proposed optimal separable convolution will be equivalent to optimizing the internal number of groups and kernel sizes to achieve the computational objective (measured in FLOPs[1]) while satisfying the volumetric RF condition. Formally, the objective function is defined by Equation (2) under the constraints defined by Equations (3)-(6). The solution to this optimization problem will be described in detail in Section 2.

---

[1]In this research, similar to (He et al., 2016), FLOPs are measured in number of multiply-adds.

We shall show that the proposed optimal separable convolution can be calculated at the order of $O(C^{\frac{3}{2}}KHW)$. This is at least a factor of $\sqrt{C}$ more efficient than the depth separable and spatial separable convolutions. The proposed optimal separable convolution is able to be easily generalized into an $N$-separable case, where the number of separated convolutions $N$ can be optimized further. In such a generalized case, an even lower complexity at $O(CHW \cdot \log(CK^2))$ can be achieved.

Extensive experiments are carried out to demonstrate the effectiveness of the proposed optimal separable convolution. As illustrated in Fig. 3, on the CIFAR10 dataset (Krizhevsky et al., 2009), the proposed optimal separable convolution achieves a better Pareto-frontier[2] than both conventional and depth separable convolutions using the ResNet (He et al., 2016) architecture. To demonstrate that the proposed optimal separable convolution generalizes well to other DCNN architectures, we adopt the DARTS (Liu et al., 2018) architecture by replacing the depth separable convolution with the proposed optimal separable convolution. The accuracy is improved from 97.24% to 97.67% with fewer parameters. On the ImageNet dataset (Deng et al., 2009), the proposed optimal separable convolution also achieves an improved performance. For the DARTS architecture, the proposed approach achieves 74.2% top1 accuracy with only 4.5 million parameters.

## 2 THE PROPOSED APPROACH

### 2.1 CONVOLUTION AND ITS COMPUTATIONAL COMPLEXITY

A convolutional layer takes an input tensor $B_{l-1}$ of shape $(C_{l-1}, H_{l-1}, W_{l-1})$ and produces an output tensor $B_l$ of shape $(C_l, H_l, W_l)$, where $C_*$, $H_*$, $W_*$ are input and output channels, feature heights and widths. The convolutional layer is parameterized with a convolutional kernel of shape $(C_l, C_{l-1}, K_l^H, K_l^W)$, where $K_l^*$ are the kernel sizes, and the superscript indicates whether it is aligned with the features in height or width. In this research, we take $C_* = O(C)$, $H_* = O(H)$, $W_* = O(W)$, and $K_*^{H|W} = O(K)$ for complexity analysis. Formally, we have

$$B_l(c_l, h_l, w_l) = \sum_{c_{l-1}} \sum_{k_l^H} \sum_{k_l^W} B_{l-1}(c_{l-1}, h_{l-1}, w_{l-1}) \cdot F_l(c_l, c_{l-1}, k_l^H, k_l^W), \tag{1}$$

where $h_l = h_{l-1} + k_l^H$ and $w_l = w_{l-1} + k_l^W$. Hence, the number of FLOPs (multiply-adds) for convolution is $C_l H_l W_l \cdot C_{l-1} K_l^H K_l^W = O(C^2 K^2 HW)$ and the number of parameters is $C_l C_{l-1} K_l^H K_l^W = O(C^2 K^2)$.

For a group convolution, we have $g$ convolutions with kernels of shape $(C_l/g, C_{l-1}/g, K_l^H, K_l^W)$. Hence, it has $O(C^2 K^2 HW/g)$ FLOPs and $O(C^2 K^2/g)$ parameters, where $g$ is the number of groups. A depth-wise convolution is equivalent to a group convolution with $g = C_* = C$. A point-wise convolution is a $1 \times 1$ convolution. A depth separable convolution is composed of a depth-wise convolution and a point-wise convolution. A spatial separable convolution replaces a $K \times K$ kernel with $K \times 1$ and $1 \times K$ kernels. Different types of convolutions are summarized in Table 1. From this table, their FLOPs and number of parameters can be easily verified.

### 2.2 RETHINKING CONVOLUTION AND THE VOLUMETRIC RECEPTIVE FIELD CONDITION

Separable convolution has been proven to be efficient in reducing the computational demand in convolution. However, existing approaches including both depth separable and spatial separable convolutions follow an ad hoc design. They are able to reduce the computational cost to some extent but will not normally achieve an optimal separation. In this research, we shall design an efficient convolution operator achieving the computational objective by optimal design of its internal hyperparameters. The resulting operator is called *optimal separable convolution*.

One difficulty is that if we do not pose any restriction to a separable convolution, optimizing the FLOPs target will resulting in a separable convolution being equivalent to a degenerated channel scaling operator[3]. Hence, we propose the following *volumetric receptive field condition*.

---

[2]In multi-objective optimization, a Pareto-frontier is the set of parameterizations (allocations) that are all Pareto-optimal. An allocation is Pareto-optimal if there is no alternative allocation where improvement can be made to one participant's well-being without sacrificing any other's. Here, Pareto-frontier represents the curve of the accuracies we are able to achieve for different FLOPs/#Params.

[3]From Table 1, let $g = C$ and $K = 1$, a convolution will have $C$ parameters and $CHW$ FLOPs. This is in fact a channel scaling operator. Composition of such operators is not meaningful because the composition itself is equivalent to a single channel scaling operator.

As illustrated in Fig. 1a, the *receptive field* (RF) of a convolution is defined to be the region in the input space that a particular CNN's feature is affected by (Lindeberg, 2013). We define the *channel RF* to be the channels that affect CNN's output and define the *volumetric RF* to be the Cartesian product of the RF and channel RF of this convolution. The volumetric RF of a convolution is actually the volume in the input space that affects CNN's output. The volumetric RF condition requires that a properly decomposed separable convolution maintains (at least) the same volumetric RF as the original convolution before decomposition. Hence, the proposed optimal separable convolution will be equivalent to optimizing its internal parameters while satisfying the volumetric RF condition.

## 2.3 OPTIMAL SEPARABLE CONVOLUTION

In this section, we discuss the case of two-separable convolution. We present the discussion informally to gain intuition into the proposed approach. In the next section, we shall provide a formal proof. Suppose that the shape of the original convolutional kernel is $(C_{out}, C_{in}, K^H, K^W)$, where $C_{in}, C_{out}$ are the input and output channels, and $(K^H, K^W)$ is the kernel size. Let $C_1 = C_{in}$, and $C_3 = C_{out}$. For the proposed optimal separable convolution, we optimize the FLOPs as computational objective while maintaining the original convolution's volumetric RF. Formally, the computational demand of the proposed separable convolution is

$$f(g_1, g_2, C_2, K_*^{H|W}) = \frac{C_2 C_1 K_1^H K_1^W HW}{g_1} + \frac{C_3 C_2 K_2^H K_2^W HW}{g_2} \quad (2)$$

In order to satisfy the volumetric RF condition, the following three conditions need to be satisfied[4]:

$$K_1^H + K_2^H - 1 = K^H \qquad \text{(Receptive Field Condition)} \quad (3)$$

$$K_1^W + K_2^W - 1 = K^W \quad (4)$$

$$g_1 \cdot g_2 \leq C_2 \qquad \text{(Channel Condition)} \quad (5)$$

$$\min(C_l, C_{l+1}) \geq g_l \qquad \text{(Group Convolution Condition)} \quad (6)$$

We have three sets of parameters: the number of groups $g_1$, $g_2$, the internal channel size $C_2$, and the internal kernel sizes $K_*^{H|W}$. In this research, we shall assume that the internal channel size $C_2$ is in an order of $O(C)$ and is preset according to a given policy. Otherwise, $g_1 = g_2 = C_2 = 1$ will be a trivial solution. This could lead the separable convolution to be over-simplified and not applicable in practice. Typical policies of presetting $C_2$ include $C_2 = \min(C_1, C_3)$ (normal architecture), $C_2 = (C_1 + C_3)/2$ (linear architecture), $C_2 = \max(C_1, C_3)/4$ (bottleneck architecture (He et al., 2016)), or $C_2 = 4\min(C_1, C_3)$ (inverted residual architecture (Sandler et al., 2018)).

The proposed problem is a constrained optimization. It is usually hard to solve it directly . However, we shall show that the optimal solution shall be $g_1, g_2 \sim \sqrt{C}$. For large channel sizes, the optimal solution is usually an interior point in the solution space rather than on the boundary. Let $K_*^{H|W}$ be constants, by substituting $g_2 = C_2/g_1$[5] and setting $f'(g_1) = 0$, one can derive that

$$g_1 = \sqrt{\frac{C_1 C_2 K_1^H K_1^W}{C_3 K_2^H K_2^W}} \sim \sqrt{C}, \quad (7)$$

and

$$\min_{g_1} f(g_1) = 2 \cdot \sqrt{C_1 C_2 C_3 K_1^H K_1^W K_2^H K_2^W} HW \quad (8)$$

$$= O(C^{\frac{3}{2}} KHW)$$

if we set $K_1^{H|W} = K^{H|W}$ and $K_2^{H|W} = 1$.

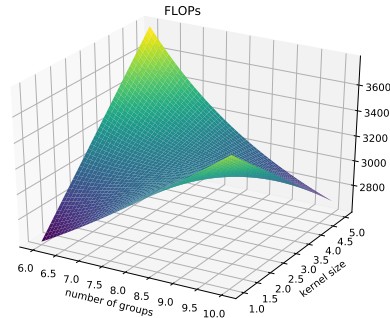

Figure 2: Given channels $C_1 = C_2 = C_3 = 64$, and kernel sizes $K^H = K^W = 5$ in Equation (2), by setting $f'(g_1) = 0$, $f'(K_1) = 0$. The solution $g_1 = 8$, $K_1 = 3$ is a saddle point.

One interesting fact is that we can optimize the internal number of groups $g_1$, $g_2$ and internal kernel sizes $K_*^{H|W}$ simultaneously. For simplicity, we assume that kernel sizes aligned in height and

---

[4]The channel condition (5) $g_1 \cdot g_2 \leq C_2 \Leftrightarrow \frac{C_1}{g_1} \cdot \frac{C_2}{g_2} \geq C_1$ means the product $\frac{C_1}{g_1} \cdot \frac{C_2}{g_2}$ needs to occupy each node in the input channel $C_1 = C_{in}$ to maintain the volumetric receptive field. This is further explained for the channel condition general case (13) in Section 2.4.

[5]It is trivial to verify that, for any solution $(g_1, g_2)$ with $g_1 \cdot g_2 < C_2$, $(g_1, \tilde{g}_2 = C_2/g_1 > g_2)$ shall be another feasible solution with a smaller FLOPs target. Hence, the optimal solution must satisfy $g_1 \cdot g_2 = C_2$.

width are equal. By setting $f'(g_1) = 0$ and $f'(K_1) = 0$, one can derive that $g_1$ is the same as Equation (7) and $K_1 = K_2 = \frac{K+1}{2}$, substituting them into Equation (8), one can get $f(g_1, K_1) = O(C^{\frac{3}{2}}K^2HW)$. This results in a higher complexity than $O(C^{\frac{3}{2}}KHW)$. In fact, the solution to $f'(g_1) = 0$ and $f'(K_1) = 0$ is a saddle point. As illustrated in Fig. 2, given the input channel $C_1 = 64$ and output channel $C_3 = 64$, kernel size $(K^H, K^W) = (5, 5)$, we take $C_2 = \min(C_1, C_3) = 64$. By setting $f'(g_1) = 0$, $f'(K_1) = 0$, the solution $g_1 = \sqrt{64} = 8$, $K_1 = \frac{5+1}{2} = 3$ is a saddle point.

## 2.4 Optimal Separable Convolution (General Case)

In this section, we shall generalize the proposed optimal separable convolution from $N = 2$ to an optimal $N$ and shall provide a formal proof. Suppose that the shape of the original convolutional kernel is $(C_{out}, C_{in}, K^H, K^W)$. Let $C_1 = C_{in}$, and $C_{N+1} = C_{out}$ ($C_{N+1} = C_3$ for $N = 2$). The computational demand of an $N$-separable convolution is

$$f(\{g_*\}, \{K_*^{H|W}\}) = \frac{C_2 C_1 K_1^H K_1^W HW}{g_1} + \cdots + \frac{C_{N+1} C_N K_N^H K_N^W HW}{g_N} \tag{9}$$

For ease of analysis, we first introduce the notation *channels per group* $n_l = \frac{C_l}{g_l}$, which simply means: channels per group $\times$ number of groups = the number of channels. Then, we have

$$f(\{n_*\}, \{K_*^{H|W}\}) = C_2 n_1 K_1^H K_1^W HW + \cdots + C_{N+1} n_N K_N^H K_N^W HW \tag{10}$$

satisfying the volumetric RF condition

$$K_1^H + K_2^H + \cdots = K^H + (N - 1) \qquad (Receptive\ Field\ Condition) \tag{11}$$

$$K_1^W + K_2^W + \cdots = K^W + (N - 1) \tag{12}$$

$$n_1 \cdots n_N \geq C_1 \Leftrightarrow g_1 \cdots g_N \leq C_2 \cdots C_N \qquad (Channel\ Condition) \tag{13}$$

$$n_l \geq \max(1, \frac{C_{l+1}}{C_l}) \Leftrightarrow g_l \leq \min(C_l, C_{l+1}) \qquad (Group\ Convolution\ Condition) \tag{14}$$

We keep both notations $g_l$ and $n_l$. This is because, for the channel condition, it is intuitive that $n_1 \cdots n_N \geq C_1$ means that the product of $n_1 \cdots n_N$ needs to occupy each node in the input channel $C_1 = C_{in}$. This is equivalent to the less intuitive condition $g_1 \cdots g_N \leq C_2 \cdots C_N$. Similarly, for the group convolution condition, $g_l \leq \min(C_l, C_{l+1})$ means the number of groups can not exceed the input and output channels of this group convolution, while $n_l \geq \max(1, \frac{C_{l+1}}{C_l})$ is less intuitive.

For Equation (10), apply an arithmetic-geometric mean inequality, we can get

$$f(\{n_*\}, \{K_*^{H|W}\}) \geq N \sqrt[N]{\frac{C_1 C_2^2 \cdots C_N^2 C_{N+1} K_1^H \cdots K_N^H K_1^W \cdots K_N^W}{g_1 \cdots g_N}} HW \tag{15}$$

$$\geq N \sqrt[N]{C_1 \cdots C_{N+1} K_1^H \cdots K_N^H K_1^W \cdots K_N^W} HW \tag{16}$$

The equality holds if and only if $C_2 n_1 K_1^H K_1^W = \cdots = C_{N+1} n_N K_N^H K_N^W$. Let $n_l = \beta_l n_1$, where $\beta_l = \frac{C_2 K_1^H K_1^W}{C_{l+1} K_l^H K_l^W}$. Let $\beta = \Pi \beta_i$, we can solve $n_1 = \sqrt[N]{\frac{C_1}{\beta}} = \frac{\sqrt[N]{\Pi C_i \Pi K_i^H \Pi K_i^W}}{C_2 K_1^H K_1^W}$ and

$$n_l = \frac{\sqrt[N]{\Pi_{i=1}^{N+1} C_i \Pi_{i=1}^N K_i^H \Pi_{i=1}^N K_i^W}}{C_{l+1} K_l^H K_l^W} \sim \sqrt[N]{C}. \tag{17}$$

Note that the inequality (16) holds for arbitrary $K_*^{H|W}$. We need to further optimize $K_*^{H|W}$. From the arithmetic-geometric mean inequality again, we can get $K_1^H \cdots K_N^H \leq (\frac{K_1^H + \cdots + K_N^H}{N})^N = (\frac{K^H + N - 1}{N})^N$ and the equality holds if and only if $K_1^H = \cdots = K_N^H = \frac{K^H + N - 1}{N}$. However, we want the inequality reversed, instead of finding the maximum of this product, we expect to find its minimum. This still gives us a hint, the maximum is achieved when the internal kernel sizes are as even as possible, so the minimum should be achieved when the internal kernel sizes are as diverse as possible. In the extreme case, one of the internal kernel sizes should take $K^H$ and all the rest takes 1. A formal proof of this claim can be derived. Hence, we have

$$f(\{n_*\}, \{K_*^{H|W}\}) \geq N \sqrt[N]{C_1 \cdots C_{N+1} K^H K^W} HW = O(NC^{1+\frac{1}{N}} K^{\frac{2}{N}} HW). \tag{18}$$

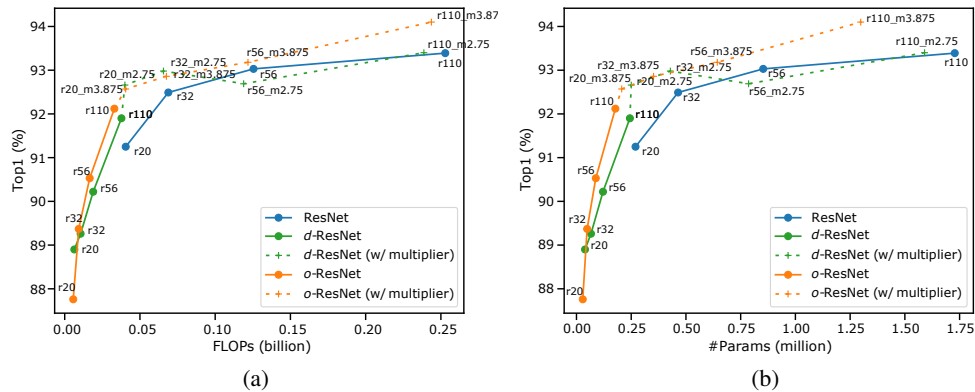

(a)                                                        (b)

Figure 3: Experimental results on CIFAR10 for the ResNet architecture (best viewed in color). The proposed optimal separable convolution (o-ResNet) achieves improved (a) accuracy-FLOPs and (b) accuracy-#Params Pareto-frontiers than both the conventional (ResNet) and depth separable (d-ResNet) convolutions.

Table 2: Experimental results on CIFAR10 for DARTS. The proposed optimal separable convolution (o-DARTS) generalizes well to the DARTS architecture, and achieves improved accuracy with approximately the same FLOPs and fewer parameters. DARTS uses depth separable convolution and an optional d- is prefixed.

| Net Arch | FLOPs | #Params | Accuracy | Error Rate |
|---|---|---|---|---|
|  | (billion) | (million) | (%) | (%) |
| (d-)DARTS (Liu et al., 2018) | 0.528 | 3.35 | 97.24% | 2.76% |
| o-DARTS | 0.572 | **3.25** | **97.67%** | **2.33%** |
| P-DARTS (Chen et al., 2019) | 0.532 | 3.43 | 97.50% | 2.50% |
| PC-DARTS (Xu et al., 2019) | 0.557 | 3.63 | 97.43% | 2.57% |
| GOLD-DARTS (Bi et al., 2020) | 0.546 | 3.67 | 97.47% | 2.53% |

It can be verified that, for $N = 2$, Equation (8) and Equation (18) match with the same complexity. By setting $f'(N) = 0$, we can derive that $N = \log(CK^2)$, and

$$\min f(\{n_*\}, \{K_*^{H|W}\}) = eCHW \cdot \log(CK^2) = O(CHW \cdot \log(CK^2)) \tag{19}$$

where $e = 2.71828...$ is the natural logarithm constant.

The proposed optimal separable convolution can have a spatial separable configuration: a single kernel takes $(K^H, K^W)$ or two kernels take $(K^H, 1)$ and $(1, K^W)$. Besides, the proposed optimal separable convolution allows using a mask of the internal number of groups and solve for an $M$-separable sub-problem $(M < N)$. Details are discussed in Appendix A, where the implementation of the proposed optimal separable convolution is also presented in Algorithm 1.

## 3 EXPERIMENTAL RESULTS

In this section, we carry out extensive experiments on benchmark datasets to demonstrate the effectiveness of the proposed optimal separable convolution scheme. In the proposed experiments, we use a prefix $d$- or $o$- to indicate that the conventional or depth separable convolutions in the baseline networks are replaced with depth separable ($dsep$) or the proposed optimal separable ($osep$) convolutions. In this research, we set the number of separated convolutions $N = 2$. The details of the training settings for the proposed experiments are described in Appendix B.

### 3.1 EXPERIMENTAL RESULTS ON CIFAR10

CIFAR10 (Krizhevsky et al., 2009) is a dataset consist of 50,000 training images and 10,000 testing images. These images are with a resolution of $32 \times 32$ and are categorized into 10 object classes. In the proposed experiments, we use ResNet (He et al., 2016) as baselines and replace the conventional convolutions in ResNet with $dsep$ and $osep$ convolutions, resulting in $d$-ResNet and $o$-ResNet.

The proposed $osep$ scheme can significantly reduce FLOPs/#Params. In Section 2, we proved that this reduction factor can be $\sqrt{C}K$ in theory[6]. As illustrated by the *solid* lines in Fig. 3 (a) and (b), the orange *solid* curve lies in a region with significantly smaller $x$-values than the blue *solid* curve.

---

[6]For optimal separable, $\sqrt{C}K = \frac{C^2K^2HW}{C^{3/2}KHW}$. For depth separable, $\frac{1}{1/K^2+1/C} = \frac{C^2K^2HW}{CHW(C+K^2)} < K^2$.

This indicates that $o$-ResNet shall have significantly smaller FLOPs and fewer parameters than the ResNet baseline. For example, $o$-ResNet*110* has even lower FLOPs (0.033 billion vs 0.041 billion) and fewer parameters (0.177 million vs 0.270 million) than ResNet*20*, yet with noticeable higher accuracy (92.12% vs 91.25%). This demonstrates that the proposed $osep$ scheme could significantly reduce both computational cost and number of parameters for conventional convolutions. For $dsep$, this reduction factor is $\frac{1}{1/K^2+1/C}$, which is bounded by $K^2$. For $3 \times 3$ kernels, this reduction can be at most 9. Whereas for the proposed $osep$ scheme, no such bounds exist. The advantage of the proposed $osep$ scheme over $dsep$ is illustrated in Fig. 3 (a) and (b) by the orange and green *solid* curves. From which, we can see the proposed $osep$ scheme is more efficient[7] with smaller $x$-values.

The proposed $o$-ResNets can have 8x-16x smaller FLOPs and 10x-18x fewer parameters than the ResNet baselines in the proposed experiments. For fair comparisons, we introduce the channel multiplier in order to approximately match the FLOPs. We use the suffix "`_m<multiplier>`"[8] to indicate the channel multiplier. Note that FLOPs/#Params is proportional to $channel\_multiplier^{3/2}$ for $osep$. As illustrated in Fig. 3, from which we can see, the proposed optimal separable convolution scheme is much more efficient than conventional convolutions. The orange curve, including both solid and dashed parts, achieved a better accuracy-FLOPs Pareto-frontier than the blue curve. It is worth noting that even under the same FLOPs, the number of $o$-ResNet parameters is also smaller than that of ResNet by a large margin. This could result in a more regularized network with fewer parameters to prevent over-fitting and possibly contribute to the final performance. In Fig. 3, we also present the $d$-ResNet curves in *dashed* green by replacing the conventional convolutions with depth separable convolutions. As can be seen, $d$-ResNet achieves good accuracy-FLOPs balances for small networks (e.g. $d$-ResNet20 and $d$-ResNet32), but performs comparable or no better than conventional convolutions for large ones (e.g. $d$-ResNet56 and $d$-ResNet110). In summary, the proposed optimal separable convolution achieves better accuracy-FLOPs and accuracy-#Params Pareto-frontiers than both conventional and depth separable convolutions.

To demonstrate that the proposed $osep$ scheme generalizes well to other DCNN architectures, we adopt the DARTS (V2) (Liu et al., 2018) network as the baseline. The DARTS evaluation network has 20 cells and 36 initial channels, we increase the initial channels to 42 to match the FLOPs. By replacing the $dsep$ convolutions in DARTS with the proposed $osep$ convolutions, as illustrated in Table 2, the resulting $o$-DARTS improved the accuracy from 97.24% to 97.67% with fewer parameters (3.25 million vs 3.35 million). It is worth noting that it is very hard to significantly improve the DARTS search space. In Table 2, we also include three variants of DARTS, i.e. P-DARTS (Chen et al., 2019), PC-DARTS (Xu et al., 2019), and GOLD-DARTS (Bi et al., 2020), with more advanced search strategies for comparison. As can be seen, $o$-DARTS even achieved higher accuracies than these advanced network architectures.

**Ablation Studies** We carry out ablation studies on the effects of the internal BatchNorms and non-linearities, and the spatial separable configuration. We conclude that internal BatchNorms and non-linearities have no effects on the results yet introduce extra computation and parameters, while the spatial separable configuration leads to a slightly worse performance. Hence, they are not adopted in this research. Details are presented in Table 5 with discussion in Appendix C.

## 3.2 EXPERIMENTAL RESULTS ON IMAGENET

We evaluate the proposed optimal separable convolution scheme on the benchmark ImageNet (Deng et al., 2009) dataset, which contains 1.28 million training images and 50,000 testing images.

### 3.2.1 IMAGENET40

Because carrying out experiments directly on the ImageNet dataset can be resource- and time-consuming, we resized all the images into $40 \times 40$ pixels. A $32 \times 32$ patch is randomly cropped and a random horizontal flip with a probability of 0.5 is applied before feeding into the network. No extra data augmentation strategies are used. The baseline ResNet architecture is a modified version of that used on the CIFAR10 dataset, except that the channel sizes are set to be $4\times$ larger, the features are

---

[7]Because there is an overhead and the channels are relatively small (e.g. 16) for CIFAR10 models, the advantage margin is noticeable but is not remarkable. This margin of advantage can be bigger for larger channels.

[8]We usually omit this channel multiplier suffix for simplicity when it is clear from the context that we are comparing different schemes under the same FLOPs/#Params and there are no confusions.

Table 3: Experimental results on ImageNet40 for the ResNet architecture. The proposed optimal separable convolution (*o*-ResNet) achieves 4-5% performance gain over the ResNet baseline.

| Net Arch | Channel Multiplier | FLOPs (billion) | #Params (million) | Accuracy (%) | Error Rate (%) |
|---|---|---|---|---|---|
| ResNet20 | - | 0.162 | 4.58 | 40.28 | 59.72 |
| *o*-ResNet20 | 5.375 | 0.160 | 5.13 | **44.94** | **55.06** |
| ResNet32 | - | 0.275 | 7.68 | 42.98 | 57.02 |
| *o*-ResNet32 | 5.75 | 0.278 | 7.78 | **47.88** | **52.12** |
| ResNet56 | - | 0.502 | 13.88 | 44.93 | 55.07 |
| *o*-ResNet56 | 6.0 | 0.497 | **12.55** | **49.97** | **50.03** |
| ResNet110 | - | 1.012 | 27.83 | 46.74 | 53.26 |
| *o*-ResNet110 | 6.25 | 1.027 | **23.79** | **50.72** | **49.28** |

Table 4: Experimental results on full ImageNet for the DARTS architecture. The proposed *o*-DARTS achieves 74.2% top1 accuracy with only 4.5 million parameters.

| Net Arch | FLOPs (billion) | #Params (million) | Top1 (%) | Top1 Error (%) | Top5 (%) | Top5 Error (%) |
|---|---|---|---|---|---|---|
| (*d*-)DARTS (Liu et al., 2018) | 0.530 | 4.72 | 73.3% | 26.7% | 91.3% | 8.7% |
| *o*-DARTS | 0.554 | **4.50** | **74.2%** | **25.8%** | **91.9%** | **8.1%** |

calculated on scales of [16, 8, 4], and the last fully-connected (FC) layer outputs 1000 categories for classification. We make this modification because the ImageNet dataset has significantly more training samples than the CIFAR10 dataset. Experimental results are illustrated in Table 3, as can be seen, by substituting conventional convolutions with the proposed optimal separable convolutions, the resulting *o*-ResNet achieved 4-5% (e.g. 49.97% vs 44.93% for 56-layer and 50.72% vs 46.74% for 110-layer) performance gains comparing against the ResNet baselines. This demonstrates that the proposed optimal separable convolution scheme is much more efficient. For *o*-ResNet56 and *o*-ResNet110, they also have fewer parameters that could contribute to a more regularized model. For *o*-ResNet20 and *o*-ResNet32, they have slightly more parameters because the last FC layer accounts for a great portion of overhead for 1000 classes.

### 3.2.2 FULL IMAGENET

Similar to the experiments on CIFAR10, we replace the $dsep$ convolutions in the DARTS (V2) network with the proposed $osep$ convolutions to demonstrate that the proposed approach is able to generalize to other network architectures. The experiment is carried out on the full ImageNet dataset. The DARTS evaluation network has 14 cells and 48 initial channels, we increase the initial channel size to 56 to match the FLOPs. The resulting network is called *o*-DARTS. Experimental results are illustrated in Table 4. It can be seen that, with fewer parameters (4.50 million vs 4.72 million), the proposed *o*-DARTS network achieved higher accuracies in both top1 (74.2% vs 73.3%) and top5 (91.9% vs 91.3%) accuracies than the DARTS baseline. This indicates that the proposed $osep$ is able to achieve better accuracy-FLOPs and accuracy-#Params balances than $dsep$ convolutions.

It is worth noting that in the proposed experiments, we adopt ResNet and DARTS as the baselines because these are two most well-known architectures. In practice, one may simply replace the conventional or depth separable convolutions with the proposed optimal separable convolutions in a DCNN to reduce the computation and model parameters. By increasing the channel sizes, better-performing models can be expected. The proposed optimal separable convolution achieved improved accuracy-FLOPs and accuracy-#Params Pareto-frontiers than both conventional and depth separable convolutions. Hence, one can either match the accuracy to get a smaller model with reduced computation and model parameters, or match the FLOPs to get a better-performing model.

## 4 CONCLUSIONS

In this paper, we have presented a novel scheme called *optimal separable convolution* to improve the computational efficiency in convolution. Conventional convolution took a costly complexity at $O(C^2 K^2 HW)$. The proposed optimal separable convolution scheme is able to achieve

its complexity at $O(C^{\frac{3}{2}}KHW)$, which is even lower than that of depth separable convolution at $O(CHW \cdot (C + K^2))$. Hence, the proposed optimal separable convolution *has the full potential to replace the usage of depth separable convolutions in a DCNN*. Examples include but are not limited to the ResNet and DARTS architectures. The proposed optimal separable convolution also has a spatial separable configuration. A generalized $N$-separable case can achieve better performance at $O(CHW \cdot \log(CK^2))$.

Another potential impact of the proposed optimal separable convolution is for the AutoML community. The proposed novel operator is able to increase the neural architecture search space. In a multi-objective optimization formulation, where both accuracy and FLOPs are optimized, we expect a more efficient network architecture can be discovered in the future using the proposed optimal separable convolution operator.

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

---

**Algorithm 1** The Algorithm for Optimal Separable Convolution

---

**Input:** Input channel $C_1 = C_{in}$, output channel $C_{N+1} = C_{out}$, kernel size $(K^H, K^W)$, number of separated convolutions $N$

**Optional Input**: internal kernel sizes (optional, preset), internal number of groups (optional, masked values), spatial separable (True or False)

**Output:** internal channel sizes $C_2, \cdots, C_N$, internal kernel sizes $K_1^{H|W}, \cdots, K_N^{H|W}$, internal number of groups $g_1, \cdots, g_N$

Calculate internal channel sizes $C_2, \cdots, C_N$ as $\min(C_{in}, C_{out})$, $\max(C_{in}, C_{out})/4$, or $4\min(C_{in}, C_{out})$, etc. according to a preset policy.

**if** internal kernel sizes $K_1^{H|W}, \cdots, K_N^{H|W}$ are not given **then**

    **if** spatial separable **then**

        Set $K_{\lfloor N/2 \rfloor}^H = K^H$, $K_{\lfloor N/2+1 \rfloor}^W = K^W$ and all other internal kernel sizes to 1.

    **else**

        Set $K_{\lfloor N/2 \rfloor}^{H|W} = K^{H|W}$ and all other internal kernel sizes to 1.

    **end if**

**end if**

Calculate internal channels per group $n_l$ according to $n_l = \frac{\sqrt[N]{\Pi_{i=1}^{N+1} C_i \Pi_{i=1}^{N} K_i^H \Pi_{i=1}^{N} K_i^W}}{C_{l+1} K_l^H K_l^W}$.

Let $g_l = \min(\lceil C_l/n_l \rceil, C_l, C_{l+1})$. If $C_l/n_l < 1$ or $C_l/n_l > \min(C_l, C_{l+1})$ for certain $l$, re-optimize $g_l$ with a masked number of groups by pre-setting $g_l = 1$ for $l \in \{l : C_l/n_l < 1\}$, $g_l = \min(C_l, C_{l+1})$ for $l \in \{l : C_l/n_l > \min(C_l, C_{l+1})\}$.

                      $\triangleright$ Because $n_l \sim \sqrt[N]{C}$, for large channel sizes, we rarely need to re-optimize.

**Return** $C_2, \cdots, C_N$; $K_1^{H|W}, \cdots, K_N^{H|W}$; $g_1, \cdots, g_N$

---

# A  ALGORITHMIC DETAILS OF THE PROPOSED OPTIMAL SEPARABLE CONVOLUTION

For the proposed optimal separable convolution, one of the internal kernel sizes can take $K^{H|W}$, while the rest takes 1. In this research, we simply select the middle kernel size as $(K^H, K^W)$. In a spatial separable configuration, we select the middle two to have their kernel sizes as $(K^H, 1)$ and $(1, K^W)$. It is worth noting that all these configurations have the same FLOPs. Unlike the spatial separable convolution where the spatial separable configuration is able to reduce the complexity from $K^2$ to $2K$. This is because the complexity has already been reduced to $O(K)$ for the proposed optimal separable convolution. Another interesting property of the proposed optimal separable convolution is that it prefers large kernel sizes over small ones.

For the proposed optimal separable convolution, we are able to preset the internal convolutional kernel sizes according to a custom policy, and optimize the internal number of groups only. Furthermore, we are able to preset a portion of the internal number of groups to certain values, and optimize only the remaining internal number of groups. Suppose that the internal channel and kernel sizes are given. Without loss of generality, we assume that $g_{M+1}, \cdots, g_N$ are preset. The proposed optimal separable problem will be an $M$-separable convolution sub-problem ($M < N$):

$$f(\{n_*\}, \{K_*^{H|W}\}) = C_2 n_1 K_1^H K_1^W HW + \cdots + C_{M+1} n_M K_M^H K_M^W HW + const \quad (20)$$

satisfying the volumetric RF condition

$$K_1^H + \cdots + K_M^H = K^H + const \qquad (Receptive\ Field\ Condition)$$

$$(21)$$

$$K_1^W + \cdots + K_M^W = K^W + const \qquad (22)$$

$$n_1 \cdots n_M \geq \frac{C_1}{n_{M+1} \cdots n_N} \Leftrightarrow g_1 \cdots g_M \leq \frac{C_2 \cdots C_N}{g_{M+1} \cdots g_N} \qquad (Channel\ Condition)$$

$$(23)$$

$$n_l \geq \max(1, \frac{C_{l+1}}{C_l}) \Leftrightarrow g_l \leq \min(C_l, C_{l+1}) \quad (Group\ Convolution\ Condition)$$

$$(24)$$

This $M$-separable sub-problem can be solved by the same algorithm. A detailed implementation of the proposed optimal separable convolution is described by Algorithm 1.

Table 5: Experimental results on CIFAR10 for the ResNet architecture with ablation studies of internal BN and non-linearity and spatial separable configuration.

| Net Arch | Channel Multiplier | FLOPs (billion) | #Params (million) | Accuracy (%) | Internal BN and Non-linearity (%) | Spatial Separable (%) |
|---|---|---|---|---|---|---|
| ResNet20 | - | 0.04055 | 0.270 | 91.25 | - | - |
| *o*-ResNet20 | 3.875 | 0.04054 | **0.206** | **92.89** | **92.74** | **92.32** |
| ResNet32 | - | 0.06886 | 0.464 | 92.49 | - | - |
| *o*-ResNet32 | 3.875 | 0.06760 | **0.352** | **93.18** | **93.22** | **92.88** |
| ResNet56 | - | 0.12548 | 0.853 | 93.03 | - | - |
| *o*-ResNet56 | 3.875 | 0.12180 | **0.643** | **93.32** | **93.42** | **92.93** |
| ResNet110 | - | 0.25289 | 1.728 | 93.39 | - | - |
| *o*-ResNet110 | 3.875 | 0.24370 | **1.298** | **94.35** | **94.21** | **93.96** |

## B  TRAINING SETTINGS

**Experiments on CIFAR10 for the ResNet architecture**   The images are padded with 4 pixels and randomly cropped into $32 \times 32$ to feed into the network. A random horizontal flip with a probability of 0.5 is also applied. All the networks are trained with a standard SGD optimizer for 200 epochs. The initial learning rate is set to 0.1, with a decay of 0.1 at the 100 and 150 epochs. The batch size is 128. A weight decay of 0.0001 and a momentum of 0.9 are used.

**Experiments on CIFAR10 for the DARTS architecture**   We follow the same training settings in (Liu et al., 2018): the network is trained with a standard SGD optimizer for 600 epochs with a batch size of 96. The initial learning rate is set to 0.025 with a cosine learning rate scheduler. A weight decay of 0.0003 and a momentum of 0.9 are used. Additional enhancements include cutout, path dropout of probability 0.2, and auxiliary towers with weight 0.4.

**Experiments on ImageNet40 for the ResNet architecture**   Each network is trained with a standard SGD optimizer for 20 epochs with the initial learning rate set to 0.1, and a decay of 0.1 at the 10 and 15 epochs. The batch size is 256, the weight decay is 0.0001 and the momentum is 0.9.

**Experiments on full ImageNet for the DARTS architecture**   We follow the training settings in (Chen et al., 2019) for multi-GPU training: the images are random resized crop into $224 \times 224$ patches with a random scale in [0.08, 1.0] and a random aspect ratio in [0.75, 1.33]. Random horizontal flip and color jitter are also applied. The network is trained from scratch for 250 epochs with batch size 1024 on 8 GPUs. An SGD optimizer with an initial learning rate of 0.5, a momentum of 0.9, and a weight decay of 3e-5. The learning rate is decayed linearly after each epoch. Additional enhancements include label smoothing with weight 0.1 and auxiliary towers with weight 0.4.

## C  ABLATION STUDIES

**Internal BatchNorm and Non-linearity**   For a DCNN, it is generally a good practice to add a BatchNorm (BN) (Ioffe & Szegedy, 2015) and a non-linearity after each convolution. For the proposed optimal separable convolution, we wonder if it is still necessary to add such a BN and a non-linearity after each of the internal separated convolutions. Experimental results are illustrated in Table 5. Comparing the "Internal BN and Non-linearity" column against the "Accuracy" column, we are able to conclude that with or without internal BN and non-linearity, similar results with only statistical variances can be generated. This is reasonable because the network has already been regularized by outer BN and non-linearity layers from the macro architecture. Internal ones shall offer little to no additional improvements. Because internal BN and non-linearity could introduce extra computation and parameters, in the proposed research, we shall not use internal BN and non-linearity.

**Spatial Separable**   Another variation of the proposed optimal separable convolution scheme is the spatial separable configuration. For Equation (16), the optimal solution is achieved when one of

Table 6: Experimental results on CIFAR10 for the ResNet with inference time on Windows 10 Intel CPU i5-8250.

| Net Arch | Channel Multiplier | FLOPs (billion) | #Params (million) | Accuracy (%) | Inference Time (s) |
|---|---|---|---|---|---|
| ResNet20 | - | 0.04055 | 0.270 | 91.25 | 0.0310 |
| $o$-ResNet20 | - | 0.00567 | 0.028 | 87.76 | 0.0276 |
| $o$-ResNet20 | 3.875 | 0.04054 | 0.206 | 92.89 | 0.0468 |
| $d$-ResNet20 | - | 0.00639 | 0.039 | 88.90 | 0.0312 |
| $d$-ResNet20 | 2.75 | 0.0400 | 0.250 | 92.66 | 0.0468 |
| ResNet32 | - | 0.06886 | 0.464 | 92.49 | 0.0469 |
| $o$-ResNet32 | - | 0.00931 | 0.048 | 89.37 | 0.0624 |
| $o$-ResNet32 | 3.875 | 0.06760 | 0.352 | 93.18 | 0.0937 |
| $d$-ResNet32 | - | 0.01057 | 0.066 | 89.26 | 0.0625 |
| $d$-ResNet32 | 2.75 | 0.06565 | 0.429 | 92.98 | 0.1154 |
| ResNet56 | - | 0.12548 | 0.853 | 93.03 | 0.0938 |
| $o$-ResNet56 | - | 0.01660 | 0.088 | 90.53 | 0.0948 |
| $o$-ResNet56 | 3.875 | 0.12180 | 0.643 | 93.32 | 0.1562 |
| $d$-ResNet56 | - | 0.01893 | 0.121 | 90.22 | 0.1094 |
| $d$-ResNet56 | 2.75 | 0.11890 | 0.786 | 92.69 | 0.1875 |
| ResNet110 | - | 0.25289 | 1.728 | 93.39 | 0.1563 |
| $o$-ResNet110 | - | 0.03300 | 0.177 | 92.12 | 0.1885 |
| $o$-ResNet110 | 3.875 | 0.24370 | 1.298 | 94.35 | 0.3216 |
| $d$-ResNet110 | - | 0.03770 | 0.244 | 91.90 | 0.1910 |
| $d$-ResNet110 | 2.75 | 0.23870 | 1.590 | 93.40 | 0.3462 |

the internal kernel sizes takes $K^{H|W}$ and all the rest takes 1. It does not matter which one of the internal kernel sizes takes $K^{H|W}$. Hence, we have this spatial separable variant: a single kernel takes $(K^H, K^W)$ or two kernels take $(K^H, 1)$ and $(1, K^W)$. The detailed implementation is illustrated in Algorithm 1. While spatial separable or not affects neither the FLOPs nor the number of parameters for the proposed optimal separable convolution, the results could be slightly different. As illustrated by the column "Spatial Separable" in Table 5, the spatial separable configuration leads to slightly worse performances. The reason might be that spatial separation fuses horizontal and vertical features separately, which could be less efficient than fusing them simultaneously.

## D  INFERENCE TIME FOR THE PROPOSED OPTIMAL SEPARABLE CONVOLUTION

FLOPs measures the best possible theoretical speed we are able to achieve. In this Section, we further report the wall-clock inference time of the proposed optimal separable convolution scheme. The inference time is measured on a laptop computer with Windows 10 operating system and Intel i5-8250 CPU, which we use to simulate a mobile platform. The results are illustrated in Table 6. As can be seen, for ResNet20, $o$-ResNet20_m3.875, and $d$-ResNet20_m2.75, they have a similar FLOPs ($\approx$0.0405 billion), yet $o$-ResNet20_m3.875 and $d$-ResNet20_m2.75 take a slightly longer inference time (0.0468s vs 0.0310s). This is because current implementation of grouped convolution in PyTorch is not optimized. From Table 6, we can also conclude that under the same FLOPs, $o$-ResNet has slightly better performance than $d$-ResNet (e.g. $o$-ResNet32 takes 0.0937s while $d$-ResNet32 takes 0.1154s). This might because the proposed optimal separable convolution scheme usually has smaller "groups" hyper-parameters than depth separable convolution. Similar conclusions can also be drawn for deeper ResNet32, ResNet56, and ResNet110 networks.

# E  RELATED WORK

There have been many previous works aiming at reducing the amount of computation in convolution. Historically, researchers apply Fast Fourier Transform (FFT) (Nussbaumer, 1981; Quarteroni et al., 2010) to implement convolution. For 1D convolution, FFT reduces the number of computations for $H$ points from $O(H^2)$ to $O(H \log H)$. For 2D convolution, FFT-2D reduces the computational complexity from $O(HW \cdot K^2)$ to $O(HW \cdot (\log H + \log W))$ (Podlozhnyuk, 2007). Hence, it can be easily concluded that FFT gains great speed up for large convolutional kernels. For small convolutional kernels ($K << H$ or $W$), a direct application is often still cheaper. Researchers also explore low rank approximation (Jaderberg et al., 2014; Ioannou et al., 2015) to implement convolutions. However, most of the existing methods obtain moderate efficiency improvements, and they usually require a pre-trained model and mainly focus on network pruning and compression. In recent state-of-the-art deep CNN models, several heuristics are adopted to reduce the heavy computation in convolution. For example, in (He et al., 2016), the authors use a bottleneck structure. Yet in (Sandler et al., 2018), the authors adopt an inverted bottleneck structure. Such heuristics may require further ad hoc design to work in practice, however, they are not solid and shall become less convincing.

Among various implementations of convolution, separable convolution has been proven to be more efficient in reducing the computational demand. Depth separable convolution is explored extensively in modern DCNNs (Howard et al., 2017; Sandler et al., 2018; Howard et al., 2019; Liu et al., 2018; Tan & Le, 2019). It reduces the computational cost of a conventional convolution from $O(C^2 K^2 HW)$ to $O(CHW \cdot (C + K^2))$. However, the proposed optimal separable convolution is even more efficient than depth separable convolution. It can be calculated at $O(C^{\frac{3}{2}} KHW)$ and has the full potential to replace the usage of depth separable convolutions. A second advantage of the proposed optimal separable convolution is that it can be applied to fully connected layers if we view them as $1 \times 1$ convolutional layers, whereas depth separable convolution cannot. Further, depth separable convolution requires the middle channel size to be equal to the input channel size, whereas for the proposed optimal separable convolution, the middle channel size can be freely set.

Spatial separable convolution was originally developed to speed up image processing operations. For example, a Sobel kernel is a $3 \times 3$ kernel and can be written as $(1, 2, 1)^T \cdot (-1, 0, 1)$. Spatial separable will require 6 instead of 9 parameters while doing the same operation. Spatial separable convolution is also adopted in the design of modern DCNNs. For example, in (Szegedy et al., 2016), the authors introduce spatial separation to the GoogLeNet (Szegedy et al., 2015) architecture. For the proposed optimal separable convolution, there is also a spatial separable configuration.

In the body of literature, separable convolution is also referred to *factorized convolution* or *convolution decomposition*. In this research, the proposed scheme is called optimal separable convolution following the naming conventions of depth and spatial separable convolutions.

