# OpenReview forum: "Rethinking Convolution: Towards an Optimal Efficiency"
_ICLR.cc/2021/Conference — Reject_

### Official Review · AnonReviewer2 · 2020-10-28
**A novel efficient separable convolution operator**

**Rating:** 6
**Confidence:** 3

**Review:**

**Summary**

This paper proposes a novel type of convolution called optimal separable convolution. Compared with existing separable convolutions like depth separable and spatial separable convolutions, the authors design a scheme to achieve an optimal separation. To prevent the proposed convolution from being degenerated, the authors define the volumetric receptive field to be the volume in the input space that affects CNN’s output. The volumetric RF condition requires that a properly decomposed separable convolution maintains the same volumetric RF as the original convolution before decomposition.

**Strengths**

- Contributions clearly stated and validated.
- Comprehensive mathematical proof seems reasonable.
- Ablation experimental results to show the effectiveness of their method.

**Weaknesses**

- The idea seems a little bit incremental in that it is a straightforward combination of group convolution and depthwise separable convolution.
- Some crucial ablation study/experimental results are missing.

**Clarity**

- The paper is well organized and easy to read.

**Comments**

- To show the efficiency of the proposed convolution, the authors are suggested to present a runtime comparison with existing separable convolutions. I understand the FLOPS results have shown the efficiency of the proposed optimal separable convolution, but it's still necessary to show the actual running times.
- The volumetric receptive field (RF) condition seems reasonable. However, the authors don't provide any ablation study on the volumetric RF condition. Say, what if removing this condition?
- The author obtains the optimal value of three sets of parameters like the number of groups, the internal channel size, and the internal kernel size.  It’s better to present some ablation studies on these parameters. These comparisons will be more powerful to demonstrate that the obtained optimal values can lead to the best result.
- This work seems an incremental version of group convolution and depthwise separable convolution. Could the authors give more discussion on this concern?
- The authors only provide experimental results on the classification task. It's interesting to see if this proposed convolution can be applied to other tasks, like segmentation.

**After Rebuttal**
I appreciate that the authors partly answered my questions and conducted experiments to show the runtime. After reading through their rebuttal and the other reviews, I will keep my original rating.

---

> ### Author Response · Authors · 2020-11-17
> **Comment for Reviewer #2**
>
> We thank the reviewer for the recognition and detailed comments.
>
> 1) “The idea seems a little bit incremental in that it is a straightforward combination of group convolution and depthwise separable convolution.”
> “This work seems an incremental version of group convolution and depthwise separable convolution. Could the authors give more discussion on this concern?”
>
> Enabling convolution separable has been proven to be an efficient way to reduce the computational complexity. However, existing separable convolutions, including depth and spatial separable convolutions, follow an ad hoc design and are NOT principled. They are able to reduce the computational cost to some degree but normally will not achieve an optimal separation. As pointed out by reviewer #1, in this research, we proposed a novel and “a principled way of designing convolution to minimize FLOPs or parameter counts without resorting to black-box optimization or algorithms of sorts.” “The key insight brought by the paper is to keep the volumetric receptive field constant.” “In contrast to existing efforts in designing efficient CNNs (e.g., NAS, pruning, quantization), this paper explores a complimentary yet relative under-explored direction, i.e., optimally design the convolutional operator.” Hence, the idea from this research leads an “under-explored direction”, instead of “a little bit incremental”.
>
> From an implementation level perspective of view, for Eq. (2), depth separable convolution heuristically takes g1=C, and g2=1. While for the proposed optimal separable convolution, the values of g1 and g2 are automatically and optimally set with Eq. (17) by minimizing the FLOPs target described by Eq. (2).
>
> From technical details perspective of view, the proposed optimal separable convolution is not a straightforward combination of group convolution and depth-wise separable convolution. Besides the channel shuffle operation mentioned in the global comment. We further need to carefully apply the group convolution. For example, the current implementation of group convolution requires groups to be divisible by both in channels and out channels. However, for the proposed optimal separable convolution, we only require g_l <= min(C_l, C_{l+1}) (Eq. (6)). (We did not include these details in this paper because we wish the readers could focus on the high-level ideas when reading, we will open-source our codes so that readers can take a reference if they want to know every detail.)
>
> 2) “Some crucial ablation study/experimental results are missing.”
> “The author obtains the optimal value of three sets of parameters like the number of groups, the internal channel size, and the internal kernel size. It’s better to present some ablation studies on these parameters.”
>
> Parameters like the number of groups, the channel sizes, and the kernel sizes are in fact hyper-parameters of a convolution. It is one of the central research topics of the AutoML framework to learn to set their values. Ablation studies based on a grid search might not be feasible and may be hard to obtain solid conclusions. We believe that it is better to leave exploring their values to the AutoML community than performing additional ablation studies by us.
>
> 3) “the authors are suggested to present a runtime comparison with existing separable convolutions.”
>
> We responded this concern in the global comment since it is also asked by the other reviewers.
>
> 4) “what if removing this condition (volumetric receptive field condition)?”
>
> The volumetric receptive field condition enforce the fusion of channel information for better classification accuracies. Almost all DCNNs are designed following this intuition. We do not think it is a good idea to remove this condition.
>
> Further, the volumetric receptive field condition is proposed to prevent the separable convolution from degenerated.  As explained in footnote 3, without this condition, optimizing the FLOPs target will result in a separable convolution that is equivalent to a degenerated channel scaling operator. This should be avoided.
>
> 5) “It's interesting to see if this proposed convolution can be applied to the other tasks, like segmentation.”
>
> We understand that the reviewer would like to see more results beyond classification, e.g. detection and segmentation. However, due to the limited computational resources and highly demanding experimentation time, the experimental results are presented only on the image classification tasks in this research. We consider to carry out additional experiments in the future.

---

### Official Review · AnonReviewer4 · 2020-10-28
**RETHINKING CONVOLUTION: TOWARDS AN OPTIMAL EFFICIENCY**

**Rating:** 6
**Confidence:** 3

**Review:**

## Summary
The paper presents a new convolution structure which tries to achieve a better balance between the efficiency and accuracy. The proposed approach is well motivated and theoretically proved. Reasonable experiments have been provided to validate the proposed algorithm.

## Pros
1. The paper is well presented and the motivation of the paper is clear.
2. The proposed convolution structure has theoretical small FLOPs and well justified based on the proof.
3. Reasonable experiments have been reported to valiate the the performance gain over the baselines.

## Cons
1. Besides from the FLOPs, is it possible to provide the computational cost for the proposed algorithm, e.g., including the inference speed in the experiments like Table 3. From the engineering implementation, the proposed structure may not be hardware-friendly.
2. For the experiments on Full Imagenet, what about the experiments for the comparison with the resnet baseline. Also, I would suggest to include the comparison with the baseline with depthwise convolution.

## Reasons for the rating
The exploration of the structure of the convolution is challenging but important to the community. The discussion of the convolution based on balance of the efficiency and effectiveness is meaningful. Although the experiments do not cover all of my concerns, I would rate it as marginally above the acceptance threshold.

## Suggestions
Please provde the the comparison of inference speed for the proposed structure. Also, it would be better to report the baseline with depthwise convolution.

---

> ### Author Response · Authors · 2020-11-17
> **Comment for Reviewer #4**
>
> We thank the reviewer for the recognition and detailed comments.
>
> 1) “Besides from the FLOPs, is it possible to provide the computational cost for the proposed algorithm”
>
> We responded this concern in the global comment since it is also asked by the other reviewers.
>
> 2) “For the experiments on Full Imagenet, what about the experiments for the comparison with the resnet baseline.”
>
> We agree with the reviewer that more experimental results on the full ImageNet dataset will be more convincing. However, due to the limited computational resources and the demanding experimentation time, we conducted the experiments on the reduced ImageNet40 dataset where the images are resized into 40x40 pixels. We have plans to carry out more experiments on the full ImageNet dataset in the future.
>
> In this research, we achieve consistent improvement for experiments on the ResNet and DARTS network architectures and on both CIFAR10 and ImageNet datasets. This suggests that the proposed optimal separable convolution is more efficient than both conventional and depth/spatial separable convolutions and our conclusion is solid and convincing.

---

### Official Review · AnonReviewer1 · 2020-10-28

**Rating:** 6
**Confidence:** 4

**Review:**

### Summary

This paper proposes a novel analysis for optimal separable convolution considering the number of parameters and FLOPs. The idea is to constraint the input information consumed stationery throughout the optimization of the parameters for separable convolutions. More specifically, this paper proposes the notion of volumetric receptive field and by holding it constant throughout optimization, one can arrive at a constrained optimization problem for solving the parameters for the optimal separable convolution. Empirical results of replacing common convolution with optimal convolution in various modern CNNs have demonstrated the effectiveness of the proposed optimal separable convolution.

### Reasons for score

I like the idea a lot. This paper provides a principled way of designing convolution to minimize FLOPs or parameter counts without resorting to black-box optimization or algorithms of sorts. The key insight brought by the paper is to keep the volumetric receptive field constant, which seems reasonable for me. With such an observation, solving for optimal separable convolution now becomes an optimization problem that can be solved efficiently. Empirical results on modern CNNs have shown the effectiveness of this approach.

### Strengths

- A novel and principled approach to design separable convolution, which is of critical importance. In contrast to existing efforts in designing efficient CNNs (e.g., NAS, pruning, quantization), this paper explores a complimentary yet relative under-explored direction, i.e., optimally design the convolutional operator. I can imagine how NAS and the proposed work be combined to lead to even better results in future work.

- Good empirical results by simply replacing old convolution operators with the proposed one.

- The analysis is easy to follow and generally agreeable to read.


### Weaknesses

- It would be interesting to see results on wall-clock time in addition to FLOPs and # Params. With that said, this is not a deal-breaker. While it might be the case that the proposed convolution operator falls short compared to the existing ones given the hardware and software implementation we have now, I still think this work would be a great motivation for hardware and software research to look into.

- It would be interesting to see results for commonly adopted CNNs on ImageNet, e.g., ResNet-50 and/or MobileNetV2. Again, this is good to have, but not a deal-breaker from my perspective.


### Post rebuttal

I appreciate the authors' efforts in conducting experiments to show the latency results. After reading through the rebuttal and the reviews from other reviewers, I would like to down-grade my score by 1.
Specifically, I agree with R3 and R4 that it would be better if experimental results are done for MobileNets/EfficientNets to empirically demonstrate the effectiveness of the optimal convolution. Those networks present strong baselines and it would be more convincing if optimal convolution indeed outperforms those. With that said, I agree with the authors that the DARTS experiments have shown that the optimal convolution can be better than depth-wise separable convolutions. As a result, I still recommend acceptance for this paper.

---

> ### Author Response · Authors · 2020-11-17
> **Comment for Reviewer #1**
>
> We thank the reviewer for the recognition and detailed comments.
>
> 1) “It would be interesting to see results on wall-clock time in addition to FLOPs and # Params.”
>
> We responded this concern in the global comment since it is also asked by the other reviewers.
>
> 2) “It would be interesting to see results for commonly adopted CNNs on ImageNet, e.g., ResNet-50 and/or MobileNetV2. Again, this is good to have, but not a deal-breaker from my perspective.”
>
> We agree with the reviewer that more experimental results on other DCNNs will be more convincing. However, due to the limited computational resources and the demanding experimentation time, we have to select only the most representative ones. We have plans to carry out more experiments on other popular network architectures in the future. In this research, we achieve  consistent improvement for experiments on the ResNet and DARTS network architectures and on both CIFAR10 and ImageNet datasets. This suggests that the proposed optimal separable convolution is more efficient than both conventional and depth/spatial separable convolutions and our conclusion is solid and convincing.

---

### Official Review · AnonReviewer3 · 2020-10-29
**Interesting ideas, strong assumptions,  relatively weak results**

**Rating:** 5
**Confidence:** 4

**Review:**

This paper proposes a new type of separable convolution to improve ConvNet efficiency. Based on a few assumptions (receptive field condition, channel condition, group conv condition), it mathematically calculate the “optimal” configurations for separable convolutions. Experiments are mostly done on CIFAR and ImageNet.

======== Strengths

1). Comprehensive study on important separable conv building blocks (e.g. Table 1 is quite informative)
2). The idea of split the groups in both stages of separable convs is somewhat new and interesting.


======== Weaknesses

1). My first concern is about the unrealistic assumptions. For example, Eq (5) “channel condition” requires g1 * g2 = C2, which doesn’t make sense to me: there is no intuition, and most existing convs doesn’t satisfy this assumption: (1) regular conv g1=g2=1 != C2 doesn’t satisfy this; (2) spatial separable conv g1=g2=1 != C2. This assumption is critical to arrive equation (7) and (8), but is unclear where this assumption comes from.  Due to these unrealistic assumptions, the term “optimal” is also questionable.

2). Second, the CIFAR results show the new layers are not much better than others. As shown in Figure 3, the largest gain is <1%, and sometimes the o-ResNet (~88%) is slightly worse than d-ResNet (which indicates the propose layers might be not "optimal"?) The improvements on ImageNet in Table 4 seem to be promising, but as discussed in DARTS+ and other recent works, the search process of DARTS is often unstable and could potentially have high variance.

3). My another main concern is about the weak baseline. As this paper is study separable convs, it should compare to separable conv based models like MobileNet/FBNet/EfficientNet, rather than the full conv based ResNet. For example, by leveraging depthwise and seprable convs, MobileNetV3 achieves 75.2% ImageNet top-1 accuracy with 219 FLOPs, which is a much stronger baseline than the on in Table4. I highly recommend the authors to conduct their experiments on these baselines.


======== Suggestions

1). Instead of formulating it as a mathematically optimal solution based on unrealistic assumptions, I recommend the authors to conduct more empirical studies on these design choices. For example, the paper only shows the performance results of “optimal” (g1, g2) computed by equation (7), but it would be helpful to show the performance for different (g1, g2) values, and compare them with the “optimal” (g1, g2).

2). I recommend the authors to use the latest MobileNet or EfficientNet (or other separable conv based models) as baselines, and replace their separable convs with the proposed “optimal separable convs”, and compare the performance gains.

---

> ### Author Response · Authors · 2020-11-17
> **Comment for Reviewer #3**
>
> We appreciate the recognition and detailed comments from Reviewer #3, and have the following responses to address the concerns.
>
> 1) “Eq (5) ‘channel condition’ requires g1 * g2 = C2, which doesn’t make sense to me: there is no intuition, and most existing convs doesn’t satisfy this assumption”
> “Due to these unrealistic assumptions, the term ‘optimal’ is also questionable.”
>
> The channel condition Eq. (5) has an intuitive meaning of “the product of n1 · · · nN needs to occupy each node in the input channel C1 = Cin” (3rd paragraph in Section 2.4). We are sorry that we did not have an explanation in Section 2.3. In the revised version, we added footnote 4 “The channel condition (5) g1*g2 <= C2 <=> (C1/g1) * (C2/g2) >= C1 means the product (C1/g1) * (C2/g2) needs to occupy each node in the input channel C1 = Cin to maintain the volumetric receptive field. This is further explained for the channel condition general case (13) in Section 2.4.” Namely, each node in the input channel needs to affect the proposed convolution’s output (from the definition of the volumetric receptive field of a convolution). Hence, we have g1*g2 <= C2 as the channel condition “derived” from the proposed volumetric receptive field condition. We hope that this will be clear to the reviewer about the intuition of the channel condition Eq. (5).
>
> For regular/spatial separable conv, the assumption is not satisfied because they are in a non-optimal configuration (in general, g1*g2<C2). For the proposed optimal separable convolution, g1*g2=C2 is a necessary condition. In the revised version, we modified Eq. (5) to g1*g2 <= C2 to eliminate this confusion. Furthermore, we added footnote 5 “It is trivial to verify that, for any solution ( g 1 , g 2 ) with g 1 ⋅ g 2 < C 2 , ( g 1 , g 2 ˜ = C 2 / g 1 > g 2 ) shall be another feasible solution with a smaller FLOPs target. Hence, the optimal solution must satisfy g 1 ⋅ g 2 = C 2 .” to make it clear. It can be seen that, the proposed problems (2) with the constraints (3)-(6) using the channel condition Eq. (5) g1*g2=C2 or g1*g2 <= C2 are equivalent. Hence, Eq. (7) and (8) shall also be valid. The main purpose of Section 2.3 is to guide the readers to look into the “saddle-point” nature of the proposed problem. In this Section, “We present the discussion informally to gain intuition into the proposed approach” (First paragraph in Section 2.3). For a mathematical formal presentation, we hope the reviewer can look into Section 2.4 and the general case channel condition (13) (by taking N=2 for the two-separable case). After reading, the confusions shall be cleared.
>
> In summary, Eq. (5) “channel condition” is valid and sound. It is “derived” from the proposed volumetric receptive field condition, and is NOT an “unrealistic assumption”. We hope our clarification above can resolve reviewer’s concern on this condition.

---

> ### Author Response · Authors · 2020-11-17
> **Comment for Reviewer #3 (Cont)**
>
> 2) “in Figure 3, the largest gain is <1%”
> “sometimes the o-ResNet (~88%) is slightly worse than d-ResNet”
> “The improvements on ImageNet in Table 4 seem to be promising, but as discussed in DARTS+ and other recent works, the search process of DARTS is often unstable and could potentially have high variance.”
>
> First, the largest gain is 1.64% (92.89%-91.25% from Table 5, Table 5 and Fig. 3 are using the same data). We consider this improvement is substantial considering that the baseline accuracy is already above 90%. Besides, ResNet110 improves ResNet20 from 91.25% to 93.39% using 5x more computation, while the proposed o-ResNet20 improves ResNet20 from 91.25% to 92.89% using NO extra computation and with noticeable less parameters. This gain is a good one.
>
> Second, in this paper, “optimal” means achieving the minimal FLOPs target under the proposed volumetric receptive field condition. In Fig. 3, this is shown as “the orange solid curve lies in a region with significantly smaller x-values than the blue solid curve” (Second paragraph in Section 3.1). From Fig. 3, we can also conclude that “the proposed optimal separable convolution is able to achieve an improved accuracy-FLOPs and accuracy-#Params trade-offs over both conventional and depth/spatial separable convolutions.” We did NOT claim that for a given FLOPs (e.g. 0.006 million), o-ResNet has achieved the best accuracy (e.g. ~88%) (this is a bold and extreme claim actually, and is NOT true, as for each year, there are always smaller networks proposed for better accuracies). Hence, “sometimes the o-ResNet (88%) is slightly worse than d-ResNet” is a normal behavior and does not affect the fact that the orange curve solid lies above the blue solid curve in Fig. 3. Hence, the conclusion that the proposed optimal separable scheme is more efficient than depth separable convolution is solid.
>
> Third, the DARTS framework has two phases: a) the search process for a final network architecture; b) re-train the final network. There might be debates on the stability of the search process for DARTS. However, there is no doubt that the final DARTS models achieved top results (>97%) on the CIFAR10 dataset. In this research, we experiment on the final DARTS models only. The proposed experiments are independent of the search process (phase a)) and hence its stability. We get consistent improvements on both CIFAR10 and ImageNet by replacing dsep convs with the proposed osep convs. This implies that our conclusion is convincing and is not affected by the stability of the search process of DARTS.
>
> 3) “As this paper is study separable convs, it should compare to separable conv based models like MobileNet/FBNet/EfficientNet, rather than the full conv based ResNet.”
> “For example, by leveraging depthwise and seprable convs, MobileNetV3 achieves 75.2% ImageNet top-1 accuracy with 219 FLOPs, which is a much stronger baseline than the one in Table4.”
>
> The reviewer might have ignored the fact that DARTS (V2) models are constructed with depth separable convolutions of extensive usage. Hence, it falls into the same family of MobileNet/FBNet/EfficientNet using depth separable convolutions. Besides, we also carry out experiments on d-ResNet, which is also depth separable convolution based. Another family of modern DCNNs are regular convolution based, e.g. VGG/ResNet. We select ResNet and DARTS as two of the best known network architectures from each family. Experimental results on both of these two families and on both CIFAR10 and ImageNet datasets suggest that the proposed optimal separable convolution is more efficient than both conventional and depth/spatial separable convolutions. We agree with the reviewer that experiments on all MobileNet/FBNet/EfficientNet networks will be more convincing. However, due to the limited computational resources and the demanding experimentation time, we have to select only the most representative ones.
>
> As for MobileNetV3 pointed out by the reviewer, it achieves 75.2% ImageNet top-1 accuracy with 219 FLOPs and 5.4 million parameters. While in Table 4, o-DARTS achieves 74.2% ImageNet top-1 accuracy with 554 FLOPs and 4.5 million parameters. It is unfair to claim that MobileNetV3 is a much stronger baseline than o-DARTS as MobileNetV3 has its advantage in smaller FLOPs, while o-DARTS has its advantage in less #Params in this case.

---

> > ### Comment · AnonReviewer3 · 2020-11-24
> > **Improved version, but still with a few limitations.**
> >
> > I appreciate the authors detailed responses. My questions are partially answered and I am happy to upgrade my score a little bit. However, I would like to have a few more comments:
> >
> > 1. Regarding assumptions
> > Good to see the change of Eq. (5) to g1*g2 <= C2”, but I am still now sure how important is this “volumetric receptive field” condition.
> >
> > 2. Regarding baselines
> >
> > First of all, I use low-FLOPs MobileNetV3 as an example because this submission aims to optimize FLOPs, as stated by the authors: “optimal means achieving the minimal FLOPs target under the proposed volumetric receptive field condition” (also see Eq 2). If you want to compare both FLOPs/Params, there are also other depthwise-conv based models (like MnasNet-A1: 75.2% top-1 with 3.9M params and 312M flops).
> >
> > Secondly, the current d-ResNet w/ multiplier has very strange performance (Figure 3: larger FLOPs lead to lower accuracy). In DARTS, in addition to separable conv, there are many other ops (dilated conv, pooling, etc) and the search algorithms also has certain variance, which make the comparison complicated. Therefore,  it is better to compare with low-FLOPs models with depth-wise (like mobilenet or similar models) or group conv (like shufflenet or similar models).
> >
> > 3. Lastly, I also have some additional questions:
> >
> > (1) Do you shuffle the channels between the two separable convs? If not, some of the channels will have no interactions?
> > (2) Could you report the latency of your whole model instead of a single conv ops? I personally think it is fine even if it is slower than expected FLOPs efficiency, but it is still helpful to provide the latency data for readers.

---

> > > ### Author Response · Authors · 2020-11-25
> > > **Comment for Reviewer #3 for Additional Concerns**
> > >
> > > We thank the reviewer for the response.
> > >
> > > 1) Regarding assumptions Good to see the change of Eq. (5) to g1*g2 <= C2”, but I am still now sure how important is this “volumetric receptive field” condition.
> > >
> > > In the second paragraph of Section 2.2, we have “One difficulty is that if we do not pose any restriction to a separable convolution, optimizing the FLOPs target will resulting in a separable convolution being equivalent to a degenerated channel scaling operator3. Hence, we propose the following volumetric receptive field condition.” And for footnote 3, we have “From Table 1, let g = C and K = 1, a convolution will have C parameters and CHW FLOPs. This is in fact a channel scaling operator. Composition of such operators is not meaningful because the composition itself is equivalent to a single-channel scaling operator.” Hence, the volumetric receptive field condition is important for a convolution. Otherwise, optimizing the FLOPs target will become degenerated.
> > >
> > > Maintaining the volumetric receptive field is also a good practice and is followed by designing almost all modern DCNNs. For example, depth-wise convolution (O(CK^2) is not used independently in a depth separable convolution. The point-wise convolution (O(C^2)) has to follow, although the point-wise convolution is more computationally expensive than the depth-wise convolution (usually C >> K^2). In fact, the volumetric receptive field condition enforces the fusion of channel information for better classification accuracies. This is also the design principle of regular Conv2d. The Conv2d operation for DCNNs is a “multi-channel” convolution operation (Eq. (1)) rather than the signal processing “single-channel” convolution to enforce the fusion of channel information.
> > >
> > > If we remove this volumetric receptive condition, this is like removing the point-wise convolution from depth separable convolution or using single-channel convolution instead of multiple-channel Conv2d in a DCNNs. This could lead to the network over-simplified and hurt the accuracies.
> > >
> > > 2) Regarding baselines
> > >
> > > We agree with the reviewer that more experimental results on other DCNNs will be more convincing. However, due to the limited computational resources and the demanding experimentation time, we have to select only the most representative ones. We have plans to carry out more experiments on other popular network architectures in the future. In this research, we achieve consistent improvement for experiments on the ResNet and DARTS network architectures and on both CIFAR10 and ImageNet datasets. This suggests that the proposed optimal separable convolution is more efficient than both conventional and depth/spatial separable convolutions and our conclusion is solid and convincing.
> > >
> > > The proposed optimal separable convolution optimizes both FLOPs and #Params. From Table 1, we can see the FLOPs and #Params are consistent with differences on H and W only. In this research, we are to show the effectiveness of the proposed optimal separable and extensive experiments are carried out on the CIFAR10 dataset. In the future, we will carry out more experiments on the full ImageNet dataset, e.g. MobileNetV3, MnasNet, ShuffleNet, etc. However, at present, we do not have so many computational resources.
> > >
> > > For a DARTS V2 cell, except for skip connections and max pooling layers, there are 10 (5x2, each sep_conv_3x3 has two depth separable convolutions actually) depth separable convolutions and 1 dilated convolution. Hence, we may think it is depth separable based. Besides, the proposed optimal separable convolution is also applicable to strided convolutions, dilated convolutions, etc.
> > >
> > > 3) “Do you shuffle the channels between the two separable convs?”
> > > “Could you report the latency of your whole model instead of a single conv ops? I personally think it is fine even if it is slower than expected FLOPs efficiency, but it is still helpful to provide the latency data for readers.”
> > >
> > > Yes. We shuffle the channels.
> > >
> > > Yes. In the revised version, we added Table 6 and Appendix D to report the latency of the whole models instead of a single conv ops. From which we can see, under approximately the same FLOPs, the proposed optimal separable convolution has slightly better latency than depth separable convolution. But both of them are slower than regular convs, which is due to the inefficient implementation of grouped Conv2d in PyTorch.

---

> > > > ### Author Response · Authors · 2020-11-25
> > > > **Quick reference for timing**
> > > >
> > > > For a quick reference on Windows 10 Intel CPU i5-8250 for ResNet20:
> > > >
> > > > ```
> > > > Time Elapsed: 0.030991315841674805 (ResNet20, FLOPS: 0.04055 billion)
> > > > Time Elapsed: 0.027650117874145508 (o-ResNet20, FLOPs: 0.00567 billion)
> > > > Time Elapsed: 0.04686284065246582 (o-ResNet20 with channel multiplier 3.875, FLOPS: 0.04054 billion)
> > > > Time Elapsed: 0.031241416931152344 (d-ResNet20, FLOPs: 0.00639 billion)
> > > > Time Elapsed: 0.04686236381530762 (d-ResNet20 with channel multiplier 2.75, FLOPs: 0.0400 billion)
> > > > ```
> > > >
> > > > **UPDATE**:
> > > >
> > > > Table 6: Experimental results on CIFAR10 for the ResNet with inference time on Windows 10 Intel CPU
> > > > i5-8250.
> > > >
> > > > | Net Arch    | Channel    | FLOPs     | #Params   | Accuracy | Inference |
> > > > | ----------- | ---------- | --------- | --------- | -------- | --------- |
> > > > |             | Multiplier | (billion) | (million) | (%)      | Time (s)  |
> > > > | ResNet20    | -          | 0.04055   | 0.27      | 91.25    | 0.031     |
> > > > | o-ResNet20  | -          | 0.00567   | 0.028     | 87.76    | 0.0276    |
> > > > | o-ResNet20  | 3.875      | 0.04054   | 0.206     | 92.89    | 0.0468    |
> > > > | d-ResNet20  | -          | 0.00639   | 0.039     | 88.9     | 0.0312    |
> > > > | d-ResNet20  | 2.75       | 0.04      | 0.25      | 92.66    | 0.0468    |
> > > > | ResNet32    | -          | 0.06886   | 0.464     | 92.49    | 0.0469    |
> > > > | o-ResNet32  | -          | 0.00931   | 0.048     | 89.37    | 0.0624    |
> > > > | o-ResNet32  | 3.875      | 0.0676    | 0.352     | 93.18    | 0.0937    |
> > > > | d-ResNet32  | -          | 0.01057   | 0.066     | 89.26    | 0.0625    |
> > > > | d-ResNet32  | 2.75       | 0.06565   | 0.429     | 92.98    | 0.1154    |
> > > > | ResNet56    | -          | 0.12548   | 0.853     | 93.03    | 0.0938    |
> > > > | o-ResNet56  | -          | 0.0166    | 0.088     | 90.53    | 0.0948    |
> > > > | o-ResNet56  | 3.875      | 0.1218    | 0.643     | 93.32    | 0.1562    |
> > > > | d-ResNet56  | -          | 0.01893   | 0.121     | 90.22    | 0.1094    |
> > > > | d-ResNet56  | 2.75       | 0.1189    | 0.786     | 92.69    | 0.1875    |
> > > > | ResNet110   | -          | 0.25289   | 1.728     | 93.39    | 0.1563    |
> > > > | o-ResNet110 | -          | 0.033     | 0.177     | 92.12    | 0.1885    |
> > > > | o-ResNet110 | 3.875      | 0.2437    | 1.298     | 94.35    | 0.3216    |
> > > > | d-ResNet110 | -          | 0.0377    | 0.244     | 91.9     | 0.191     |
> > > > | d-ResNet110 | 2.75       | 0.2387    | 1.59      | 93.4     | 0.3462    |

---

### Author Response · Authors · 2020-11-17
**Common Comment**

For Reviewers #1, #2, #4, one common comment is on the wall-clock time of the proposed optimal separable convolution scheme. We have the following response to address the reviewers’ common concerns. For the other concerns, we address them to individual reviewers separately.

[#R1] “It would be interesting to see results on wall-clock time...”

[#R2] “Besides from the FLOPs...”

[#R4] “To show the efficiency of the proposed convolution...”

Short Answer:

First, FLOPs measure the best possible theoretical speed we are able to achieve, while wall-clock time depends on the hardware. The authors would like to thank all three reviewers for their appreciation and recognition of the theoretical efficiency for the proposed optimal separable convolution. We do understand - reviewers’ request for wall-clock time in addition to FLOPs and #Params. However, since the experiments could be carried out on different hardware, it is best to present the current results independent of specific hardware.

We hope the theoretical speed we are able to achieve  measured in FLOPs is adequate. As the proposed optimal separable convolution is “ahead” of current implementation of grouped convolution in PyTorch or TensorFlow, we implemented the proposed optimal separable scheme with the PyTorch Python API which is not optimized. An efficient implementation shall require careful tweak of the CUDNN library. As commented by Reviewer #1, we wish “this work would be a great motivation for hardware and software research to look into”.

Based on the attached testing code, one can still appreciate the effectiveness and potential impact of the proposed optimal separable convolution scheme. For a large convolution calculated on Windows 10 Intel CPU i5-8250, the following are the statistics:

```
Regular conv: 12.79s, FLOPs: 1087 billion, #Params: 132 million
Depth separable conv: 0.748s, FLOPs: 43.9 billion, #Params: 5.4 million
Optimal separable conv: 0.281s, FLOPs: 16.3 billion, #Params: 2.0 million
```

where the effectiveness of the proposed optimal separable convolution scheme is clearly demonstrated.


Long Answer:

It is not a secret that current mainstream implementations of depth separable convolution in both PyTorch and TensorFlow are slow. For example, https://github.com/pytorch/pytorch/issues/18631, or https://github.com/tensorflow/tensorflow/issues/12132. The reason for the slowness can be complex. However,one reason is that separable convolution has an intermediate output, this can introduce extra memory access cost (MACs). The other reason is that for a grouped convolution, each group will call a CUDNN subroutine and introduce an overhead. This overhead can be ignored if groups=1, but it will accumulate to a large overhead if groups=channels (e.g. 1024). The proposed optimal separable convolution is based on grouped convolutions. Hence, it inherits the same weakness as depth separable convolutions. This is also pointed out by Reviewer #1, “it might be the case that the proposed convolution operator falls short compared to the existing ones given the hardware and software implementation”.

Depth separable convolutions are not calculated efficiently on GPUs, it still of great usage on CPUs and Mobile platforms (ARM architectures). For example, depth separable convolutions are extensively used for current mainstream deep learning researches, e.g MobileNet/DARTS/EfficentNet etc. The proposed optimal separable convolution also has the same potential usages. As commented by Reviewer #1, “I can imagine how NAS and the proposed work be combined to lead to even better results in future work.”

We implemented the proposed optimal separable convolution scheme in PyTorch using existing operators (GroupedConv2d, ChannelShuffle, GroupedConv2d). It is not optimized. One obvious optimization is that the ChannelShuffle operation could be integrated into the first GroupedConv2d to avoid the extra copy of data. However, this will require to rewrite the codes in CUDNN levels. In fact, the proposed optimal separable convolution is ahead of current implementation of Conv2d in PyTorch. As commented by Reviewer #1, “I still think this work would be a great motivation for hardware and software research to look into”. We wish that our research will motivate the hardware to optimize to achieve this theoretical speedup.

**Update**:

Thanks to Reviewer #3 and also Reviewers #1/#2/#4, the following is the inference time for CIFAR-ResNet20 (Windows 10 Intel CPU i5-8250), we also added Table 6 and Appendix D for ResNet32/56/110 in the revision:

```
Time Elapsed: 0.030991315841674805 (ResNet20, FLOPS: 0.04055 billion)
Time Elapsed: 0.027650117874145508 (o-ResNet20, FLOPs: 0.00567 billion)
Time Elapsed: 0.04686284065246582 (o-ResNet20 with channel multiplier 3.875, FLOPS: 0.04054 billion)
Time Elapsed: 0.031241416931152344 (d-ResNet20, FLOPs: 0.00639 billion)
Time Elapsed: 0.04686236381530762 (d-ResNet20 with channel multiplier 2.75, FLOPs: 0.0400 billion)
```

---

> ### Author Response · Authors · 2020-11-17
> **Sample Testing Code**
>
> ```python
> from time import time
> import math
> import torch
> import torch.nn as nn
> from torchvision import models
> from mytorch.utils.viz import print_summary # print_summary prints FLOPs and #Params
>
> CHANNEL = 256*9
>
> x = torch.randn(32, CHANNEL, 16, 16)
> conv = nn.Conv2d(CHANNEL, CHANNEL, 5, padding=2, bias=False)
> __TIC = time()
> y = conv(x)
> __TOC = time()
> print('Time Elapsed: {}'.format(__TOC - __TIC))
> print(print_summary(conv, inputs=(x,)))
>
> conv2 = nn.Sequential(nn.Conv2d(CHANNEL, CHANNEL, 5, padding=2, bias=False, groups=CHANNEL),
>                       nn.Conv2d(CHANNEL, CHANNEL, 1, padding=0, bias=False))
> __TIC = time()
> y2 = conv2(x)
> __TOC = time()
> print('Time Elapsed: {}'.format(__TOC - __TIC))
> print(print_summary(conv2, inputs=(x,)))
>
> from mytorch.nn.sep_conv import OptSepConv
> # conv3 = OptSepConv(CHANNEL, CHANNEL, 5, padding=2, bias=False)
> # NOTE: the following implementation of OptSep is over-simplified and is without ChannelShuffle
> #             The general calculation of “groups” should follow Eq. (17) in the paper.
> conv3 = nn.Sequential(nn.Conv2d(CHANNEL, CHANNEL, 3, padding=1, bias=False, groups=int(math.sqrt(CHANNEL))),
>                       # ChannelShuffle should go here
>                       nn.Conv2d(CHANNEL, CHANNEL, 3, padding=1, bias=False, groups=int(math.sqrt(CHANNEL))))
> __TIC = time()
> y3 = conv3(x)
> __TOC = time()
> print('Time Elapsed: {}'.format(__TOC - __TIC))
> print(print_summary(conv3, inputs=(x,)))
> ```

---

### Decision · Program_Chairs · 2021-01-07
**Final Decision**

**Decision:**

Reject

**Comment:**

This paper introduces a novel convolution-like operator called "optimal separable convolution" which is based on minimizing number of operations given a fixed receptive field.  Authors provide further empirical results to show the effectiveness of their proposed operator.

Overall, this is a very interesting work. There is a consensus among reviewers that this work is well-motivated, novel and principled. However, reviewers have pointed to several issues that makes this a borderline paper and consequently none of the reviewers were willing to argue for the acceptance. After reading the paper, reviewers' comments and authors' response, I would summarize the main areas of improvements as follows:

1- The "optimal separable convolution" is derived theoretically using "volumetric receptive field condition". However, this condition is not discussed and motivated enough in the paper. For example, different parametrization with the same volumetric receptive field could impose very different expressive power or implicit bias. Why is this not important? Adding discussions/experiments to motivate this condition would improve the paper.

2- The derivations in Sections 2.3 and 2.4 are not well-presented and are hard to follow. I suggest authors to use the convention of having a formal Theorem statement followed by the proof. This is important since one of the main contributions of the paper is a principled derivation.

3- All reviewers were concerned with the wall-clock time. Authors responded that theoretical #FLOPs is more important because wall-clock time is hardware dependent. However, authors reported the wall-clock time using CPUs. I understand that wall-clock time is hardware dependent but that only means algorithms that can have better wall-clock time on the current hardware are more likely to be useful because there is no guarantee that the hardware would be adjusted based on one algorithm especially if the promised improvement is not large enough. Therefore, I think reporting Wall-clock time on GPUs is important which was not done here.

4- Even though authors mention several operators in Table 1, they only compare against depth separable conv in the experiments. Even based on FLOPs, the current empirical results are not very promising. For example:

a) The gap between o-ResNet (the proposed method) and d-ResNet is not significant in Fig 3. In particular when #FLOPs is low, d-ResNet and o-ResNet have similar performance.

b) In Tables 2 and 4, o-ResNet shows small improvements but uses more FLOPs. Even if authors can't exactly match #FLOPs, they should make sure that the proposed method uses less FLOPs than others not the other way around.

c) In Table 3, authors only compare to ResNet and d-ResNet is removed.

Considering the above issues, I think the paper is marginally below acceptance threshold. Given the novelty of the work, I want to encourage the authors to improve the paper by taking Reviewers' comments into account and resubmit their work.